# Mutually Exclusive Expression of Closely Related Odorant-Binding Proteins 9A and 9B in the Antenna of the Red Flour Beetle *Tribolium castaneum*

**DOI:** 10.3390/biom11101502

**Published:** 2021-10-12

**Authors:** Alice Montino, Karthi Balakrishnan, Stefan Dippel, Björn Trebels, Piotr Neumann, Ernst A. Wimmer

**Affiliations:** 1GZMB, Department of Developmental Biology, Johann-Friedrich-Blumenbach-Institute for Zoology and Anthropology, Ernst-Caspari-Haus, Georg-August-University Goettingen, Justus-von-Liebig-Weg 11, 37077 Goettingen, Germany; alice.metzger@biologie.uni-goettingen.de (A.M.); sdippel@gwdg.de (S.D.); 2Goettingen Graduate Center for Neurosciences, Biophysics, and Molecular Biosciences, Georg-August University School of Science, University of Goettingen, 37077 Goettingen, Germany; 3Department of Forest Zoology and Forest Conservation, Buesgen-Institute, Georg-August-University Goettingen, Buesgenweg 3, 37077 Goettingen, Germany; karthimrgn@gmail.com; 4Department of Biology—Animal Physiology, Philipps-University Marburg, Karl-von-Frisch-Str. 8, 35032 Marburg, Germany; trebels@biologie.uni-marburg.de; 5GZMB, Department of Molecular Structural Biology, Institute of Microbiology & Genetics, Ernst-Caspari-Haus, Georg-August-University Goettingen, Justus-von-Liebig-Weg 11, 37077 Goettingen, Germany; pneuman2@uni-goettingen.de

**Keywords:** beetle, chemosensation, Coleoptera, insect, olfaction, protein interaction

## Abstract

Olfaction is crucial for insects to find food sources, mates, and oviposition sites. One of the initial steps in olfaction is facilitated by odorant-binding proteins (OBPs) that translocate hydrophobic odorants through the aqueous olfactory sensilla lymph to the odorant receptor complexes embedded in the dendritic membrane of olfactory sensory neurons. The *Tribolium castaneum* (Coleoptera, Tenebrionidae) OBPs encoded by the gene pair *TcasOBP9A* and *TcasOBP9B* represent the closest homologs to the well-studied *Drosophila melanogaster* OBP Lush (*Dmel*OBP76a), which mediates pheromone reception. By an electroantennographic analysis, we can show that these two OBPs are not pheromone-specific but rather enhance the detection of a broad spectrum of organic volatiles. Both OBPs are expressed in the antenna but in a mutually exclusive pattern, despite their homology and gene pair character by chromosomal location. A phylogenetic analysis indicates that this gene pair arose at the base of the Cucujiformia, which dates the gene duplication event to about 200 Mio years ago. Therefore, this gene pair is not the result of a recent gene duplication event and the high sequence conservation in spite of their expression in different sensilla is potentially the result of a common function as co-OBPs.

## 1. Introduction

Insects rely heavily on odorous stimuli to find food or hosts or to recognize partners. Odorant reception occurs in chemosensory sensilla and is supported by odorant-binding proteins (OBPs), odorant receptors (ORs), sensory neuron membrane proteins, ionotropic glutamate-like receptors, gustatory receptors, and odorant degrading enzymes [1]. The antennae carry the highest density of olfactory sensilla, which represent hair-like structures that house the dendrites of the odorant receptor neurons and are filled with aqueous sensilla lymph. This lymph contains OBPs that are secreted by non-neuronal auxiliary cells [2,3]. OBPs are globular, rather small (10 to 30 kDa), water-soluble proteins (reviewed in [4] with a hydrophobic ligand-binding pocket [5]. Systematic OBP knock-downs in *D. melanogaster* indicate their necessity for correct olfactory behavioral responses and suggest a combinatorial OBP-dependent odorant recognition [6]. Moreover, *D. melanogaster* mutants for the OBP Lush (OBP76a; [7]), and an allelic variation of different OBPs in *D. melanogaster* [8] and of an OBP in the fire ant *Solenopsis invicta* [9], as well as several RNAi based experiments in mosquitoes [10,11] showed that OBPs are essential for the correct reception of different semiochemicals in these insects. Functional experiments conducted with moth pheromone receptors in heterologous expression systems [12,13,14] or in vivo using the *D. melanogaster* “empty neuron system” [15,16] revealed that the presence of the corresponding OBP (pheromone-binding protein, PBP) increases the sensitivity to the pheromone by two to three orders of magnitude (reviewed in [1]). Therefore, hydrophobic semiochemicals are believed to first interact with OBPs, which shuttle them through the aqueous sensillar lymph, to finally reach and activate the OR/Orco complex [1]. However, OBPs have also been reported to be involved in the inactivation of the odor response by removing odorants from the sensillar lymph and the receptors [1,17], which then actually enables the reactivation of the respective OR [18]. More recently, Xiao et al. [19] showed in *D. melanogaster* that many OBP-depleted sensilla have no significant changes in their odor responsiveness and Larter et al. [20] demonstrated that the deletion of the abundant OBP28a does not cause a reduction of the olfactory response but rather an increase. This indicates a buffering role of this OBP and thus presents a potential molecular mechanism of gain-control.

Comparative expression data suggest that within the classic OBPs, the antenna-binding proteins II (ABPII) subgroup has a specific role in olfaction, since all members of *Tribolium*, *Drosophila*, and *Anopheles* are highly expressed and enriched in the antennae [21]. Moreover, this group contains some of the most prominent OBPs such as *Anopheles gambiae Agam*OBP4 that forms cooperative heteromers with other OBPs [22], *Agam*OBP1 that is co-expressed with other ABPIIs (*Agam*OBP3, *Agam*OBP4, *Agam*OBP19) [23] and mediates indole detection to find blood meals [10], as well as *D. melanogaster* Lush that is involved in pheromone detection in trichoid sensilla [7,24]. Lush is at present the most thoroughly investigated OBP and has been demonstrated to bind the *Drosophila* pheromone 11-*cis*-vaccenyl acetate [25,26] as well as to other insect pheromones [27], short-chain alcohols [28,29], and phthalates [30].

The red flour beetle *Tribolium castaneum* (Herbst, Coleoptera, Tenebrionidae) is a major pest of stored products [31] and currently represents the best established coleopteran model organism [32] with a number of excellent genetic tools: environmental RNA interference (RNAi) [33,34], forward genetics-based insertional mutagenesis [35], genome editing [36], transgene-based mis-expression systems [37,38], as well as a fully annotated genome sequence [39,40,41]. These tools, its ground dwelling lifestyle, and its evolutionary position relatively far away from Dipterans and Lepidopterans dedicate *Tribolium* to investigate findings from *Drosophila* for their generality in insects. In this study, we focused on the functional and expression analysis of *TcasOBP9A* and *TcasOBP9B* which are the two *T. castaneum* OBPs most closely related to the well-studied *D. melanogaster* OBP Lush.

## 2. Material and Methods

### 2.1. Tribolium Rearing

*Tribolium castaneum* strain San Bernardino was reared on organic whole wheat flour supplemented with 5% brewer’s yeast powder in aired transparent plastic boxes at 30 °C and 40% relative humidity under a photo regime of 12 h light and 12 h dark.

### 2.2. Cloning of TcasOBP9A and TcasOBP9B

To amplify the open reading frames (ORFs) of *T. castaneum* OBP9A and OBP9B without the predicted signal peptide, PCR using Advantage2Taq polymerase (Clonetech, Mountain View, CA, USA) was performed on cDNA from total RNA, which was prepared from antennae by using the ZR Tissue & Insect RNA MicroPrep (Zymo, Irvine, CA, USA) followed by the SMART cDNA Synthesis Kit (Clonetech, Mountain View, CA, USA) and applying gene specific primers (Table 1). These PCR products were cloned into the TA Dual Promoter PCRII vector (Invitrogen, Life Technologies GmbH, Darmstadt, Germany) and confirmed by Sanger sequencing (Macrogen, Seoul, Republic of Korea).

### 2.3. RNA Interference

The templates for the bidirectional in vitro dsRNA synthesis were generated by PCR using a T7 and an SP6 primer with a T7 promoter sequence overhang. The dsRNA was then synthesized by using the T7 Megascript Kit (Ambion, Life Technologies GmbH, Darmstadt, Germany) with the respective PCR fragment as the template. After DNAse treatment and LiCl precipitation, the RNA was annealed by boiling for about 20 min cooling down to room temperature over 3 h. The formation of dsRNA was confirmed by gel electrophoresis and the concentration was adjusted to 3 µg/µL with injection buffer (1.4 mM NaCl, 0.07 mM Na_2_HPO_4_, 0.03 mM KH_2_PO_4_, 4 mM KCL, and 10% phenol red), using a NanoDrop ND-100 (NanoDrop Technologies, Inc., Wilmington, NC, USA). The mock dsRNA was made from a PCRII vector containing a 427 bp fragment of DsRed (kindly provided by G. Bucher).

For the injection of dsRNA, late pupae were mounted with double sided adhesive tape on an object slide and approximately 0.5 µL of the dsRNA solution was injected into the conjuctivum between the fourth and fifth abdominal segments using a pulled glass capillary coupled to a FemtoJet express (Eppendorf, Hamburg, Germany). The injected pupae were placed in flour filled Petri dishes and stored in an incubator at 30 °C until adult beetles emerged, which were then aged and food-deprived before their use for EAG experiments.

### 2.4. Electroantennography

To perform the electroantennography (EAG) recordings, we selected five previously identified compounds [42]: 2-hexanone (Sigma-Aldrich, food related: mold, fruity), β-ionone (ABCR, Karlsruhe, Germany, food related, ripe grain), 4,8-dimethyldecanal (TRÉCÉ Inc., Adair, OK, USA, aggregation pheromone, 4,8-DMD), (*E*)-2-heptenal (Sigma-Aldrich, food related: fatty, green), and 6-methyl-5-hepten-2-one (ABCR, Karlsruhe, Germany, food related: fruity, green). These volatile compounds were selected based on measurable EAG response values, their structural dissimilarities (Figure 1), and their biological importance to flour beetles. The selected compounds were all ≥96% pure.

EAG (Syntech, Hilversum, The Netherlands) was used to record the antennal responses of dsRed and dsRNAi-OBP injected beetles to selected volatile compounds. A detailed procedure for EAG recording and odor presentation is described in Balakrishnan et al. [42]. For initial EAG recordings, 10–15, 20–25, or 30–35 days post injection (DPI) beetles were used for antennal preparation (Appendix A). Selected volatile compounds were diluted in silicone oil M 200 (Carl Roth GmbH + Co. KG, Germany) and dilutions in logarithmic steps 10^−2^–10^−6^ (*w*/*w*) were prepared. The time intervals between stimuli were one and two minutes, respectively. Antennal EAG responses in mV were recorded with the manufacturer’s EAG program (Syntech, version 2.7, EAG 2000). The further analysis was performed as described in Trebels et al. [43]. For subsequent EAG recordings, 10–15 DPI beetles were used and for analysis, male and female data were pooled, since no consistent differences were observed (Appendix A).

### 2.5. Antennal Fluorescent In Situ Hybridization

Synthesis of digoxigenin (DIG) or biotin-labelled RNA probes was conducted as described in [44] as fragmented probes were stored at -20 °C in 50% formamide, 10% dextran sulfate, 0.2 μg/μL yeast tRNA, 0.2 μg/μL sonicated salmon sperm DNA, and 2× SSC.

Fluorescence in situ hybridization (FISH) on *T. castaneum* antennae was performed as described for *Anopheles gambiae* antennae (Karner et al., 2015) with several modifications: after fixation, the antennae were transferred into silicone molds (E4015, Sigma-Aldrich), embedded in tissue freezing medium (“Tissue-Tek^®^ O.C.T. Compound”, Science Services GmbH, München, Germany), and frozen at −20 °C for at least 10 min, followed by cutting into 50 µm sections at −23 °C on a cryotome (Cryostat CM 1950, Leica, Nussloch, Germany) resulting in longitudinally bisected antennae. Subsequently, the frozen slices were collected in cold Eppendorf tubes and washed twice for 1 min in PBS to remove the melted embedding media, followed by 10 min in 0.2 M HCl and 1 min incubation in PBS + 1% Triton X-100. Afterwards, the antennae were kept in the hybridization solution (50% formamide, 5× SSC, 1× Denhardt’s reagent, 50 µg/mL yeast RNA, 1% Tween 20, 0.1% Chaps, 5 mM EDTA, pH 8.0) for 1 to 10 days at 4 °C. The half-mounts were prehybridized at 55 °C for 5 h before adding the probes. After probe incubation for 3 days at 55 °C, the antennae were washed four times for 15 min each in 0.1× SSC at 60 °C, followed by blocking unspecific binding sites with 1% blocking reagent (Roche) for 5 h at 4 °C. For the detection of DIG-labelled probes, Fab fragments of anti-digoxigenin-AP antibodies (Roche Diagnostics Deutschland GmbH, Mannheim, Germany) were diluted 1:500 in blocking reagent, incubated for 3 days at 4 °C, washed 5 × 10 min with TBS, 0.05% Tween 20, and eventually visualized with the HNPP Fluorescent Detection Set (Roche Diagnostics) one to three hours at room temperature. Biotin-labelled probes were detected by Streptavidin-HRP conjugate (1:100, PerkinElmer, Rodgau, Germany) and visualized with the TSA™, Fluorescein Syste” or the TSA™ Plus Fluorescein System (PerkinElmer, Rodgau, Germany). Nuclei were stained with DAPI (1: 1000). Finally, antennae were washed three times for 5 min each in TBS and transferred to PBS before they were embedded in Mowiol mounting media (10% polyvinylalcohol 4–88, 20% glycerol in PBS). The embedded samples were stored at −20 °C and analyzed by confocal microscopy.

### 2.6. Microscopy and Image Processing

The fluorescent-labeled antennae samples were scanned with a Zeiss LSM780 laser scanning microscope (Carl Zeiss Microscopy GmbH, Jena, Germany) using a 405 nm, 488 nm, and 561 nm laser. Confocal image stacks were taken from single antennal segments. The confocal image stacks were converted into maximal intensity projections with AMIRA graphics software (FEI Visualization Sciences Group, Mérignac Cedex, France). The final images were arranged and labelled using Photoshop (Adobe, San José, CA, USA).

### 2.7. Phylogenetic Analysis and Interspecies Comparison

The *Dmel*Lush and *Dmel*OBP19a orthologous of all non-coleopterans were collected based on data from Flybase [45] and OrthoDB [46]), the beetle OBPs were collected from several publications, and relevant orthologs were selected based on a preliminary phylogenetic analysis with Fast Tree (v2.1.5) [47]. We chose OBPs from the hymenopteran species *Apis mellifera* [48,49], *Arpegnathos saltator* [50,51], *Linepithema humile* [52], and *Nasonia vitripennis* [53]; from the lepidopterans *Bombyx mori* [54], *Heliconius melpomene* [55], and *Danaus plexippus* [56]; from the dipterans *Glossina morsitans* [57], *Anopheles gambiae* [58,59], and *Drosophila melanogaster* [45]; from the coleopterans *Anoplophora chinensis* [60], *Anomala corpulenta* [61], *Ambrostoma quadriimpressum* [62], *Brontispa longissimi* [63]), *Colaphellus bowringi* [64], *Callosobruchus chinensis* [65], *Cylas formicarius* [66], *Dendroctonus ponderosae* [67,68], *Holotrichia oblita* [69], *Galeruca daurica* [70], *Leptinotarsa decemlineata* [71], *Rhynchophorus ferrugineus* [72], and *Tribolium castaneum* [21]; from the blattodean *Blattella germanica* [73], as well as the isopteran *Zootermopsis nevadensis* [74].

After the subtraction of the signal peptide (SignalP4.1 [75]), the sequences were aligned using MAFFT (v7.388 [76]) and the tree was constructed using RAxML (version 8.2.11 [77]) with BLOSUM62 substitution model and GAMMA correction. The robustness of the tree topology was evaluated by 100 rapid bootstrap replications. The phylogenetic tree was visualized by iTOL [78] and descriptions were added using inkscape (www.inkscape.org, last accessed on 11 October 2021).

### 2.8. Homology Modeling and In Silico Docking Experiments

Three-dimensional (3D) models of OBP9A and OBP9B (Appendix A) have been calculated using a comparative (homology) modeling approach implemented in ROSETTA [79]. Additional odorant-binding proteins (OBP4D, OBP5D, OBP5F, HoblOBP2) have been calculated by Phyre2 server [80]. The obtained 3D homology models have been subjected to a further refinement using the relax protocol running many side chain repack and minimization cycles as implemented in [81,82]). The lowest energy models have been selected for further analyses (one out of 1000 decoys).

Protein–protein docking has been performed using an approach designed for the local refinement of docked structures with ROSETTA [83]. The initial orientation of two monomers forming a dimeric arrangement has been obtained based on the superposition with a homodimeric *Drosophila* odorant-binding protein Lush complexed with butanol (PDB id: 1OOH). For each heterodimer, the most optimal dimeric model has been selected based on the lowest *Isc* score representing the energy of the interactions across the dimeric interface (one out of 1000 decoys). The analysis of the dimer interface has been performed using PISA software [84].

Docking of small molecular compounds (2-heptanal, methylheptenone, beta ionone, 4,8-dimethyldecanal, 2-hexanon) has been performed utilizing qvina and smina software (“AutoDock Vina: improving the speed and accuracy of docking with a new scoring function, efficient optimization and multithreading”, www.ncbi.nlm.nih.gov/pmc/articles/PMC3041641/last accessed on 2 April 20) for homology models of OBP9A and OBP9B used as binding proteins. Different exhaustiveness levels (50, 100, 150, 200, 250, 300, 350) defining the time spent on the search have been tried in order to decrease the probability of not finding the minimum energy decoys. During these docking calculations a set of flexible side chains have been used. For OBP9A and OBP9B models, conformations of nine side chains (residues: Leu/Met 112, Ile108, Leu51, leu56, Leu71, Phe63, Phe121, Leu122, Leu7) have been optimized during docking. Flexible side chains of the remaining homology models have been selected based on superposition with OBP9A (structurally equivalent residues). A visual analysis of the docking results has been performed using VIDA (the Free Public Domain Research License of OpenEye Scientific Software, Inc., Santa Fe, NM. http://www.eyesopen.com, last accessed on 11 October 2021). The tabular data of interactions between the docked odors and OBP molecules (OBP9A and OBP9B) have been obtained using the ncont program from the CCP4 [85] suite employing 3.9 Å distance as the threshold. Illustrations depicting the OBP–odor interactions (Appendix A) have been prepared with the LigPlot+ [86]

## 3. Results and Discussion

### 3.1. TcasOBP9A and TcasOBP9B Enhance Detection of a Broad Spectrum of Volatiles

In order to identify whether *TcasOBP9A* and *TcasOBP9B* facilitate the detection of specific odorants, we employed EAG to examine the antennal response of dsRNA-mediated RNAi-treated beetles to selected volatile compounds (with DsRed dsRNA as the mock control). Our initial EAG screening of 94 compounds [42] identified five volatile organic compounds (VOCs) to be readily detected by *T. castaneum*: 2-hexanone, β-ionone, 4,8-dimethyl-decanal, 2-heptenal, and 6-methyl-5-hepten-2-one. To identify the time window of the RNAi-mediated knock-down, we used *dsRNA TcasOBP9A*-injected female beetles from three different time points after injection for EAG recording: 10–15 DPI (days post injection), 20–25 DPI, and 30–35 DPI. This preliminary work revealed that the dsRNA-injected beetles showed highly (**/***) significant EAG response reduction at 10–15 DPI to the highest concentration (10^−2^) of all tested odors. This effect was reduced at 20–25 DPI for 4,8-dimethyl-decanal, 2-heptenal, and 6-methyl-5-hepten-2-one, and at 30–35 DPI for 4,8-dimethyl-decanal and 6-methyl-5-hepten-2-one and gone for the other odors (Appendix A), indicating that the high transcription rate of OBPs in the auxiliary cells titers out the RNAi effect over time. Therefore, we used beetles 10-15 DPI for further experiments, in which we silenced *TcasOBP9A*, *TcasOBP9B*, and both by dsRNA-mediated RNAi. In order to highlight the dose-dependent reduction of EAG responses after the RNAi treatment of the selected compounds, we used a wide dilution range of 10^−6^–10^−2^ (log_10_ dilution in silicone oil).

Unexpectedly, the EAG response to all of these structurally very diverse odorants, with substantial different binding affinities to the OBPs (Appendix A), was significantly reduced at the highest odor concentration already when *TcasOBP9B* was knocked-down by RNAi (Figure 1). The knock-down of OBP9A caused a similar, but weaker, effect on all odors, leading to a significant EAG response reduction in 2-hexanone, 2-heptenal, and 6-methyl-5-hepten-2-one. Additionally, the EAG response to the predicted best ligand for both OBPs, the food-related odor beta-ionone (Appendix A), was only weakly affected by the OBP knock-downs. Therefore, these highly abundant OBPs (Dippel et al., 2014) do not seem to provide any specificity to the olfactory process; however, they enhance the detection of diverse VOCs in an unspecific manner.

### 3.2. Mutual Exclusive Antennal Expression of TcasOBP9A and TcasOBP9B

Based on quantitative transcriptomics data, *TcasOBP9A* and *TcasOBP9B* belong to the most abundantly expressed OBPs in the *T. castaneum* antenna of both sexes [21]. However, these data do not allow one to answer the question of whether these OBPs are expressed in all or only a subset of olfactory sensilla. Therefore, we performed FISH experiments to identify the exact pattern of the antennal expression for *TcasOBP9A* and *TcasOBP9B* (Figure 2). Whereas *TcasOBP9A* is only expressed in the terminal 11th article of the antenna (Figure 2A), *TcasOBP9B* is expressed in all three olfactory responsive club articles (Figure 2B). Double FISH with *TcasOBP9B* and *TcasOrco* as a marker for olfactory sensory neurons [87] revealed the distally adjacent location of the auxiliary cells to the olfactory sensory neuron somata (Figure 2C) and thereby confirmed the arrangement already described by Roth and Willis [88]. In contrast to other insects, both the auxiliary cells as well as the olfactory sensory neuron somata are located not directly beneath the sensilla cuticle but rather more proximally within the club segments.

Double FISH with *TcasOBP9A* and *TcasOBP9B* indicates that there is no overlap of expression (Figure 2D–D’’). While *TcasOBP9A* is expressed more centrally in the terminal segment, *TcasOBP9B* is expressed radially in all three olfactory club segments. This expression pattern could correlate with the distribution of trichoid sensilla centrally in the terminal segment and basiconic sensilla radially in segments 9–11 [87]. Therefore, it is likely that these two genes are mutually exclusively expressed in auxiliary cells associated with different types of sensilla. This is somewhat surprising, as the two genes resemble a gene pair directly neighboring each other on the ninth chromosome with only a 1.5 kb intergenic region [21].

### 3.3. Phylogeny of TcasOBP9A and TcasOBP9B

By comparing OBPs only across different insect orders [21], a detailed picture of gene duplication events cannot be provided. Therefore, we carried out a phylogenetic analysis for specifically Lush-related OBPs across a variety of insect orders but also including a large set of coleopteran species (Figure 3). This analysis shows that specific *TcasOBP9A* and *TcasOBP9B* paralogs can be identified across Cucujiform beetles, which indicates that the gene duplication event must date to about 200 Mio years ago [89]. Thus, despite the close chromosomal localization and the best homology within *T. castaneum*, this is not a gene pair derived from a recent gene duplication. There was much time for those genes to gain independent expression. In this respect, it is interesting to note that *TcasOBP9A* expression is restricted to the head, whereas *TcasOBP9B* is also significantly expressed in the legs [21], which resembles the expression of the respective orthologues in the sweet potato weevil *Cylas formicarius*, *CforOBP11* and *CforOBP4* [66]. There are no obvious remnants of a lost paralog in the genome of *T. castaneum*, which would support a second, more recent gene duplication. In addition, the detailed comparison of the intron–exon structure does not support gene conversion between *TcasOBP9A* and *TcasOBP9B*.

## 4. Conclusions

Since the *TcasOBP9A* and *TcasOBP9B* gene pair is not the result of a recent gene duplication and their expression has become mutual exclusive, the reason for their close sequence homology (Figure 4 and Appendix A) must lie in a common function. For some Lush orthologs, it has been shown that they can form functional heterodimers that bind odorants better than single OBPs, e.g., AgamOBP4 [22,90] and the homologue HoblOBP4 of the scarab beetle *Holotrichia oblita* [91]. Both AgamOBP4 and HoblOBP4 can bind to a wide variety of VOCs [22,91] and *AgamOBP4* has been shown to be abundantly expressed in many sensilla and to be co-expressed with other OBPs in specific sensilla [23]. Such a function as a kind of co-OBP would also be consistent for TcasOBP9A and TcasOBP9B with their abundant expression and the enhancement of the detection of very diverse odorants. Additionally, the highest degree of sequence conservation between OBP9A and OBP9B concentrates on amino acids that form the outer surface of helices one, two, and six resulting in an almost complete conserved surface area, which could act as a docking site to interact with other proteins (depicted in Figure 4B). A future analysis of the detailed expression of more *T. castaneum* OBPs [21] will reveal potential co-expression with *TcasOBP9A* or *TcasOBP9B* and thus identify candidates for the study of heterodimer formation in this beetle.

## Figures and Tables

**Figure 1 biomolecules-11-01502-f001:**
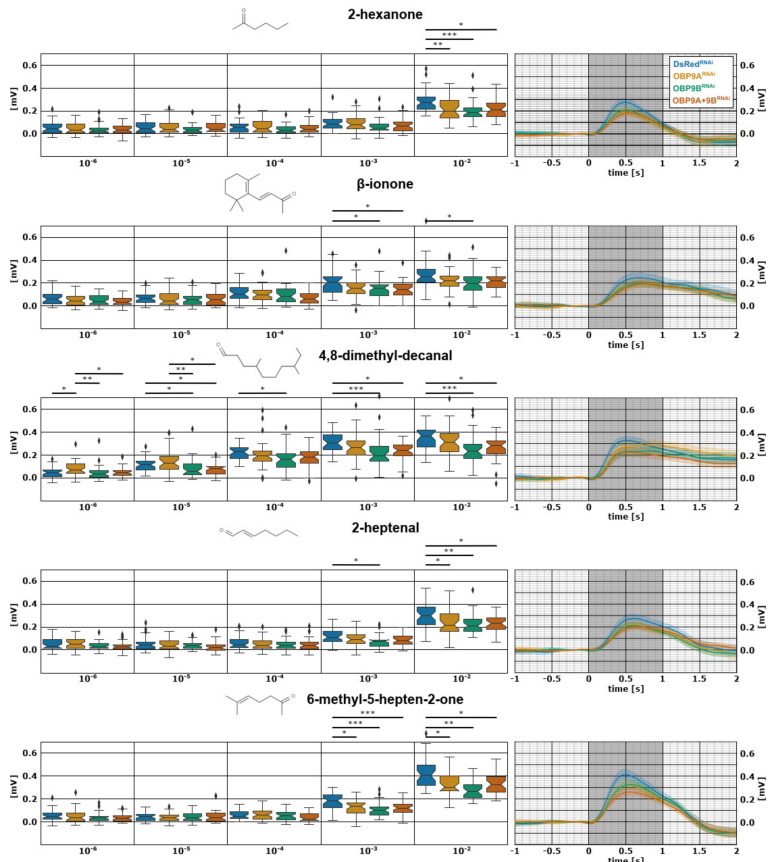
Electroantennographic (EAG) responses of twelve (N_animals_ = 12, 6 of each sex, n_replicates_ = 3) beetles to the olfactory stimulants 2-hexanone, β-ionone, 4,8-dimethyl-decanal, 2-heptenal, or 6-methyl-5-hepten-2-one, 10–15 days after dsRed (blue), OBP9A (orange), OBP9B (green), or OBP9A + OBP9B (red) dsRNA injection. On the left, box plots with whiskers representing the 5–95% percentile of the peak amplitude EAG response in mV after robust LOESS smoothing and normalization to five log_10_ dilutions (10^−2^–10^−6^) of aforementioned compounds. Statistical analysis between different dsRNAs was performed per odor and dilution by Dunn’s multiple comparison test, following initial Kruskal–Wallis test. Asterisks represent statistical significance levels (Holm corrected) of difference in median (* *p*_corr_ < 0.05, ** *p*_corr_ < 0.01, *** *p*_corr_ < 0.001). On the right side, line plots of the mean EAG response to the respective odor at a concentration of 10^−2^ after robust LOESS smoothing and normalization. Shaded areas represent the confidence interval of the median calculated by bootstrap analysis. One second odor stimulus is indicated by the dark grey box.

**Figure 2 biomolecules-11-01502-f002:**
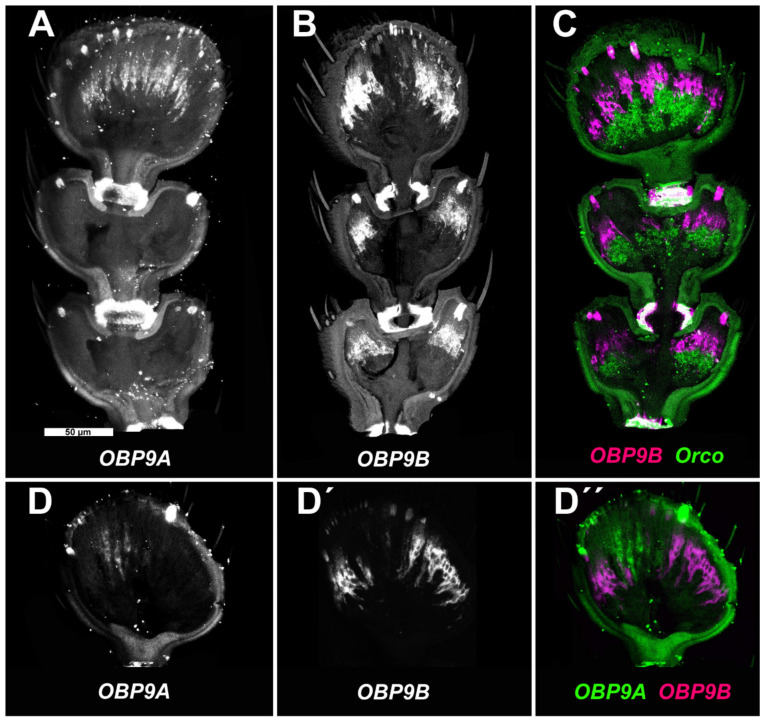
Expression of *OBP9A* and *OBP9B* in *T. castaneum* antennae. Maximal intensity projections of confocal stacks of articles 9–11 (**A**–**C**), representing the olfactory responsive club segments [87], or the terminal article 11 (**D**–**D’’**) after fluorescent in situ hybridization (FISH). (**A**) Digoxigenin-labelled probe targeting *TcasOBP9A* transcripts visualized by the HNPP/FastRed detection system for digoxigenin-labelled probes. (**B**) Biotin-labelled probe targeting *TcasOBP9B* transcripts visualized by the TSA detection system for biotin-labelled probes. (**C**) Double FISH with a digoxigenin-labelled probe targeting *TcasOrco* transcripts (green) [87] and a biotin-labelled probe targeting *TcasOBP9B* transcripts (magenta). (**D**–**D’’**) Double FISH with a digoxigenin-labelled probe targeting *TcasOBP9A* (**D**) transcripts and a biotin-labelled probe targeting *TcasOBP9B* (**D’**) transcripts. (**D’’**) overlay of (**D**) (green) and (**D’**) (magenta). Note that the strong signal at the article boarders and the sensilla bases are non-specific accumulations. Scale bar in A is 50 µm and applies to all panels.

**Figure 3 biomolecules-11-01502-f003:**
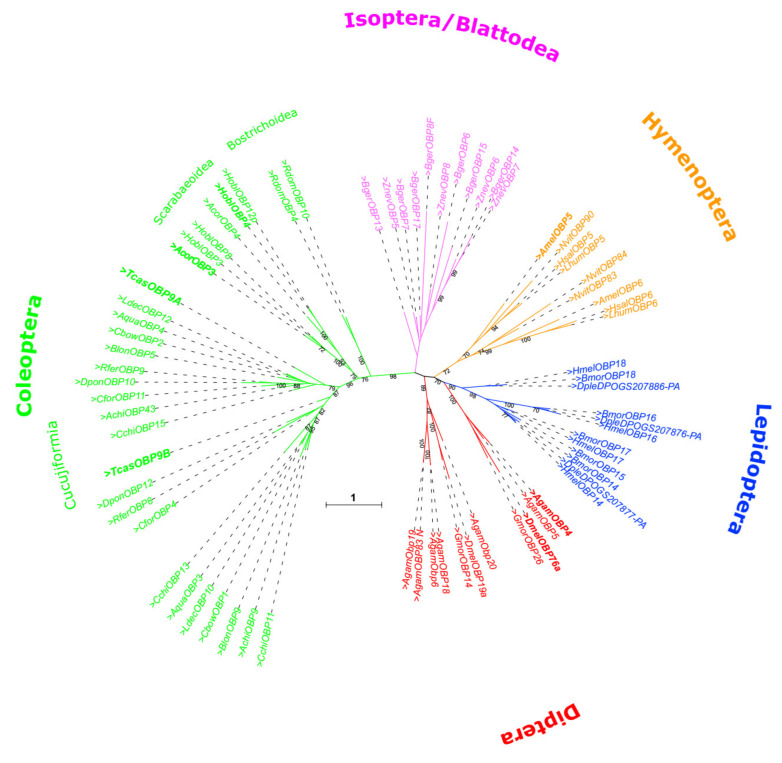
Unrooted phylogenetic tree of TcasOBP9A and TcasOBP9B orthologs from various insects (Appendix A). The gene duplication of *Tribolium* leading to the paralogous TcasOBP9A and TcasOBP9B is also present in the Curculionidae (*D. ponderosae*, *R. ferrugineus*, *C. formicarius*), Chrysomelidae (*C. bowringi*, *G. daurica*, *B. longissimi*, *A. quadriimpressum*, *C. chinensis*), and the Cerambycidae (*A. chinensis)*, but not in the Scarabaeidae (*H. oblita*, *A. corpulenta*) and the Bostrichidae (*R. dominica*). The scale bar within the tree represent 1 amino acid substitution per site. Black numbers on branches show values of 100 times the replication bootstrap analysis higher than 70.

**Figure 4 biomolecules-11-01502-f004:**
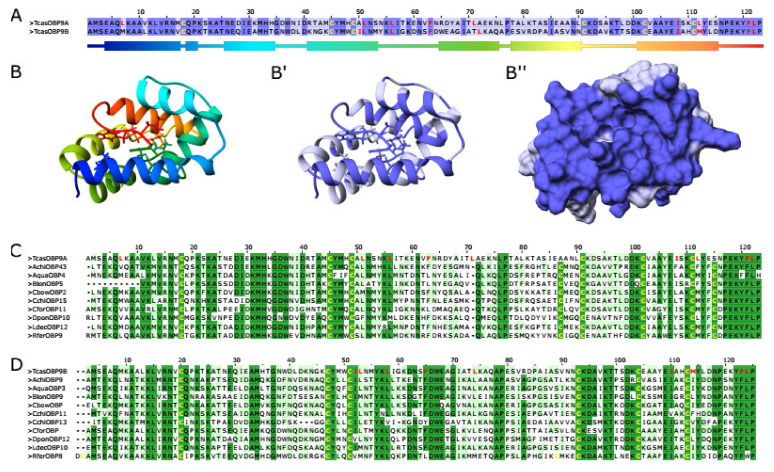
Sequence comparison of TcasOBP9A and TcasOBP9B. (**A**) Amino acid alignment of TcasOBP9A and TcasOBP9B made with MAFFT [76] and visualized with Jalview [92] highly conserved AA are highlighted in dark blue, structural relevant cysteins are yellow, and residues that flex upon odor binding are red. Below is the domain structure of OBP9A with coloring corresponding to the rainbow ribbon model in B. (**B**–**B’’**) Ribbon model of OBP9A based on modelling with ROSETTA [79] and visualized with Chimera [93] embedded in Jalview [92], the residues that are flexible upon odor binding are depicted as stick models. (**B**) Coloring as rainbow based on the AA position, (**B’**) coloring based on AA conservation between OBP9A and OBP9B (as in **A**), (**B’’**) surface model of B’. (**C**) Amino acid alignment of TcasOBP9A with OBPs from the same branch (see Figure 3). (**D**) Amino acid alignment of TcasOBP9B with OBPs from the same branch (see Figure 3).

**Table 1 biomolecules-11-01502-t001:** The primers used for dsRNA synthesis.

Forward Primer	OBP	Reverse Primer
aaaccATGGCTGCGATGTCTGAGGC	OBP9A *EcoRIrev*	TTTGAATTCTCAGGGGAGAAAGTACTTTTCAGGATTG
aaaccATGGCGATGAGTGAAGCCC	OBP9B *EcoRIrev*	TTTGAATTCTTACGGTAAGAAGTATTTCTCGGGATTATCC

## Data Availability

Not applicable.

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
