# Peer review of "Mutually Exclusive Expression of Closely Related Odorant-Binding Proteins 9A and 9B in the Antenna of the Red Flour Beetle Tribolium castaneum"

_biomolecules, 2021, doi:10.3390/biom11101502_

Round 1

Reviewer 1 Report

  1. The authors detected the EAG response at 10-15 days after dsRNA treatments. However, there were not much description on the dsRNA treatments and the authors did not show the time, dose and collecting time. According to our experience on dsRNA treatment, dsRNA treatment did not mean it will take effect after injection during the treatment.
  2. The overall structure of manuscript might regulate again and it can first analyze the bioinformation, molecular docking, and then to identify the target molecular functions.
  3. The results and discussion look like there are still incomplete and not discussion enough.
  4. It did not show the direct link between molecular function and phylogenetic analysis.

Author Response

Rebuttal Reviewer 1:

  1. The authors detected the EAG response at 10-15 days after dsRNA treatments. However, there were not much description on the dsRNA treatments and the authors did not show the time, dose and collecting time. According to our experience on dsRNA treatment, dsRNA treatment did not mean it will take effect after injection during the treatment.

What has been described for the injection in the material and methods has been the following:

For the injection of dsRNA, late pupae were mounted with double sided adhesive tape on an object slide and approximately 0.5 µL of the dsRNA solution were injected into the conjuctivum between the fourth and fifth abdominal segments., using a pulled glass capillary coupled to a FemtoJet express (Eppendorf, Hamburg, Germany). The injected pupae were placed in flour filled Petri dishes and stored in an incubator at 30°C until adult beetles emerged, which were then aged and food-deprived before their use for EAG experiments.

And: … the RNA was annealed by boiling for about 20 min cooling down to room temperature over 3 h. The formation of dsRNA was confirmed by gel electrophoresis and the concentration was adjusted to 3 µg/µL with injection buffer …

This indicates the developmental time point of collection and injection (late pupae) the dose (0.5 microl of 3microg per microl: Thus calculated 1,5 microgram). The analysis was at least ten days after injection and not during the injection. WE think this has been properly described.

2.The overall structure of manuscript might regulate again and it can first analyze the bioinformation, molecular docking, and then to identify the target molecular functions.

There are many ways to arrange the results from very diverse types of experiments. We found that the current order is the best way to present the data. Coming from what is known on the biological function to find new unexpected data on the expression and try to discuss these based on the molecular data. The proposed order would make it hard to tell a story. Therefore, we would like to stay with our order of presenting the data.

3.The results and discussion look like there are still incomplete and not discussion enough.

There are different experimental paths to follow up based on these data. However, we wanted to provide what we have so far to the scientific community. Further discussion on this would be highly speculative and therefore, we would like to stick to presenting the data with minimal speculation and leave it to further experimental analysis to provide more data, which would then allow more precise conclusions.

4.It did not show the direct link between molecular function and phylogenetic analysis.

To prove a direct link between a molecular function and an evolutionary meaning is a really hard thing to do and not very often shown. Here we present an interesting observation of the mutual exclusive expression of two OBPs that are not representing a recent gene duplication despite their close similarity both in structure and function. We are using the phylogenetic and the molecular data to build up a hypothesis that will have to be further analyzed in the future.

Reviewer 2 Report

The study was authored by Montino et al., investigated the mutual expression of closely related antenna OBP 9A and 9B of red flour beetle. Overall, the manuscript was well-written and explained all the methods were carried out in this manuscript. Specially, the introduction and methods provide important data to the readers. However, there are few major concerns to clarify and update few corrections to the improve the article. There are major and minor points were given below.

Major comments

  1. The authors were mentioned that the comparative modeling of OBP9A and OBP9B using rosetta. What are the templates were used to construct the protein models? line 235: where is the structural superimposition results of Drosophila OBP Lush with OBP9A and OBP9B? 
  2. Line 93-94: The authors must to submit the sequence comparison of between Drosophila OBP Lush between OBP9A and OBP9B as supplementary figure.
  3. Figure 3: The phylogenetic tree showed that the Drosophila OBP (Lepidoptera) protein is highly deviated and in another opposite branch from OBP9A and OBP9B (Coleoptera) clade?
  4. Figure 4: B and B' protein figures had ligands in the cavity and which is not explained in the legend file. The protein figure is not clear in the B and B'. How the conserved sites were identified and indicated in the figure B'?
  5. Supplementary figure S3. There is not residual interaction were identified and discussed in the article. The authors must to submit the protein-ligand interactions (H-bond, hydrophobic) as a table and 2D residual interaction map.
  6. Result and discussion: The result part was described well but few more important points to discuss about the figure 3, 4 and supplementary figure S3 results.

Minor comments

  1. The reference format/numbers are not corresponding to the maintext reference provided in the article.
  2. Line 26, 286, 321: Change as "OBP" insted of OPB. verify all over the manuscript and supplementary files.
  3. Line 130: Correct as "Balakrishnan et al., 2017" and verify all over the manuscript
  4. Line 131-132 and Table 1: follow the same fonts in whole manuscript.
  5. Line 154: write correctly the superscript (10 power of -2, -6).

Author Response

Rebuttal Reviewer 2:

Major comments

1.The authors were mentioned that the comparative modeling of OBP9A and OBP9B using rosetta. What are the templates were used to construct the protein models? line 235: where is the structural superimposition results of Drosophila OBP Lush with OBP9A and OBP9B? 

This information is now provided in NEW Supplementary Figure S3.

2.Line 93-94: The authors must to submit the sequence comparison of between Drosophila OBP Lush between OBP9A and OBP9B as supplementary figure.

This information is now provided in NEW Supplementary Figure S3.

3.Figure 3: The phylogenetic tree showed that the Drosophila OBP (Lepidoptera) protein is highly deviated and in another opposite branch from OBP9A and OBP9B (Coleoptera) clade?

This phylogenetic tree shows only the orthologs of TcasOBP9A and 9B and not all OBPs. Therefore it is expected that the members of the different insect orders separate. Why some dipteran OBPs are based on a week bootstrap value somewhat closer to the Lepidopteran orthologs is not of importance here.

4.Figure 4: B and B' protein figures had ligands in the cavity and which is not explained in the legend file. The protein figure is not clear in the B and B'. How the conserved sites were identified and indicated in the figure B'?

There are no ligands depicted, the AA residues which flex during ligand binding are shown: “the residues that are flexible upon odor binding are depicted as stick models!
“Conserved sites were identified by the pairwise alignment in (A) and are depicted in the same dark blue color as in the alignment. This is actually stated in the Figure legend.

5.Supplementary figure S3. There is not residual interaction were identified and discussed in the article. The authors must to submit the protein-ligand interactions (H-bond, hydrophobic) as a table and 2D residual interaction map.

This information is now provided in NEW Supplementary Figure S4 and old Supplementary Figure S3 is now Supplementary Figure S5.

6.Result and discussion: The result part was described well but few more important points to discuss about the figure 3, 4 and supplementary figure S3 results.

Further discussion on this would be highly speculative and therefore, we would like to stick to presenting the data with minimal speculation and leave it to further experimental analysis to provide more data, which would then allow more precise conclusions.

Minor comments

7.The reference format/numbers are not corresponding to the maintext reference provided in the article.

The style of referencing needs to be finally adjusted after acceptance. In the moment the list is alphabetical and just numbered without any representation of the numbers in the text.

8.Line 26, 286, 321: Change as "OBP" insted of OPB. verify all over the manuscript and supplementary files.

Thanks for spotting this. Has been corrected.

9.Line 130: Correct as "Balakrishnan et al., 2017" and verify all over the manuscript

Thanks for spotting this. Has been corrected here and at several other places throughout.

10.Line 131-132 and Table 1: follow the same fonts in whole manuscript.

Thanks for spotting this. Has been corrected.

11.Line 154: write correctly the superscript (10 power of -2, -6).

Thanks for spotting this. Has been corrected.

Round 2

Reviewer 1 Report

no other comments

Reviewer 2 Report

The authors have provided all the response and modified all the major and minor corrections in the manuscript and added new supplementary figures.

Table 1: not followed the same fonts in the revised manuscript.